# State of the Art on the Role of *Postmortem* Computed Tomography Angiography and Magnetic Resonance Imaging in the Diagnosis of Cardiac Causes of Death: A Narrative Review

Chiara Stassi [1], Cristina Mondello [1,*], Gennaro Baldino [1], Luigi Cardia [2], Patrizia Gualniera [1], Fabrizio Calapai [3], Daniela Sapienza [1], Alessio Asmundo [1] and Elvira Ventura Spagnolo [1,*]

1   Department of Biomedical and Dental Sciences and Morphofunctional Imaging, University of Messina, Via Consolare Valeria, 1, 98125 Messina, Italy; chiara_stassi@libero.it (C.S.); gennarobld@hotmail.it (G.B.); patrizia.gualniera@unime.it (P.G.); daniela.sapienza@unime.it (D.S.); aasmundo@unime.it (A.A.)
2   Department of Human Pathology in Adult and Developmental Age "Gaetano Barresi", University of Messina, Via Consolare Valeria, 1, 98125 Messina, Italy; luigicardia1@gmail.com
3   Department of Chemical, Biological, Pharmaceutical and Environmental Sciences, University of Messina, 98125 Messina, Italy; fabrizio.calapai@unime.it
*   Correspondence: mondelloc@unime.it (C.M.); eventuraspagnolo@unime.it (E.V.S.); Tel.: +39-3477062414 (C.M.); +39-3496465532 (E.V.S.)

**Abstract:** The need of a minimally invasive approach, especially in cases of cultural or religious oppositions to the internal examination of the body, has led over the years to the introduction of *postmortem* CT (PMCT) methodologies within forensic investigations for the comprehension of the cause of death in selected cases (e.g., traumatic deaths, acute hemorrhages, etc.), as well as for personal identification. The impossibility to yield clear information concerning the coronary arteries due to the lack of an active circulation to adequately distribute contrast agents has been subsequently overcome by the introduction of coronary-targeted PMCT Angiography (PMCTA), which has revealed useful in the detection of stenoses related to calcifications and/or atherosclerotic plaques, as well as in the suspicion of thrombosis. In parallel, due to the best ability to study the soft tissues, cardiac *postmortem* MR (PMMR) methodologies have been further implemented, which proved suitable for the detection and aging of infarcted areas, and for cardiomyopathies. Hence, the purpose of the present work to shed light on the state of the art concerning the value of both coronary-targeted PMCTA and PMMR in the diagnosis of coronary artery disease and/or myocardial infarction as causes of death, further evaluating their suitability as alternatives or complementary approaches to standard autopsy and histologic investigations.

**Keywords:** *postmortem* coronary angiography; *postmortem* MRI; coronary atherosclerotic disease; *postmortem* CT; cardiomyopathy

## 1. Introduction

Since the first application of Computed Tomography (CT) to *postmortem* practice in 1983, the use of Multi-Detector CT (MDCT) and Magnetic Resonance Imaging (MRI) has widely spread within the forensic field in order to evaluate their potential as either complementary approaches or alternatives to conventional autopsy for the determination of the cause of death [1–3]. Although autopsy is well established as the gold standard technique for investigating the cause of death, the use of a less invasive radiological approach—whenever appropriate—might be preferable in some circumstances, especially in cases of cultural or religious oppositions to the internal examination of the body [3–7]. Since they allow a better visualization of bone structures, gas/fluid spaces and foreign bodies, also providing an enhanced resolution thanks to 3D volume rendering techniques, *postmortem* CT (PMCT) methodologies are most suited for the investigation of traumatic

deaths, acute hemorrhages, lung parenchyma diseases, calcifications (stones, atherosclerosis), pneumothorax and pathological conditions associated with free air, as well as for personal identification [3,5,6,8–11]. Compared to PMCT, *postmortem* Magnetic Resonance (PMMR), although less used, provides useful information on the cause of death as well, proving superior in the study of soft tissues [3,5].

Nonetheless, the inadequate distribution of contrast agents due to the lack of an active circulation, excluded the possibility to yield clear information concerning the coronary arteries [1,12–14]. In order to overcome such limit, PMCT Angiography (PMCTA) has thus been introduced for its high ability to enhance the cardiovascular system visualization. A first whole-body *postmortem* angiographic approach was carried out by the Virtopsy® group [15] by using a modified heart–lung bypass machine [1,3,13] but, due to its complexity, more selective approaches were further implemented, consisting of the injection of the contrast agent in the targeted organ—the heart—before or after its removal, and subsequent scan [1,13].

The present work provides an overlook on the actual applications of cardiac PMCTA and PMMR in order to evaluate (1) their diagnostic potential in cases of cardiac death and (2) their reliance as alternative or complementary diagnostic approaches to conventional autopsy. The research was performed on PubMed and Scopus databases, using four main groups of keywords: "post mortem CT"; "post mortem CT angiography"; coronary targeted PMCT"; "post mortem MR". The articles selected included reviews, original articles, prospective studies and case reports; references from the chosen articles were also evaluated for possible inclusion. Exclusion criteria were languages different from English and unavailability.

## 2. Role of PMCTA in the Diagnosis of Coronary Artery Disease and Myocardial Infarction as Causes of Death

Ischemic heart disease represents one of the leading causes of cardiac death worldwide. The underlying impaired blood supply, usually related to coronary atherosclerosis, can lead to either myocardial infarction, chronic ischemic heart disease or sudden cardiac death [2,4,7,16,17]. Autopsic and histologic investigations actually figure as the gold standard approaches to detect either coronary calcifications and atherosclerotic plaques—and related grade of stenosis—thrombosis or infarcted areas, while the further implementation of immunohistochemistry to the forensic research is increasingly proving valuable for a better definition of myocardial ischemic lesions staging [18–20]. Although the advent of radiological methods applied to a *postmortem* setting has introduced, with good results, the possibility of their use for diagnostic purposes with a minimally invasive approach, the diagnosis of coronary artery disease (CAD) as cause of death still remains challenging. In such a context, with the implementation of angiography to conventional PMCT a few works have been produced in order to evaluate the potential of coronary-targeted PMCTA, as alternative to conventional autopsy, in the detection of atherosclerotic plaques and calcification-related stenoses, thrombosis and myocardial infarction as causes of death (Table 1).

Coronary calcifications can be easily detected, as shown in the work by Michaud et al. [2], who compared the findings of native PMCT, PMCTA, conventional autopsy and histological examination of coronary arteries and myocardium in 23 cases of suspected cardiac death. Both PMCT and PMCTA detected coronary calcifications in 18/23 cases (78%), in 12 of which an associated coronary thrombosis was also observed at autopsy. In one case on PMCT, in which no calcifications were detected, PMCTA revealed a 50–75% stenosis of one coronary artery corresponding, in histologic investigation, to an acute thrombosis related to an eroded plaque. In another case, both PMCT and PMCTA evidenced a small calcification of the left anterior descending (LAD) artery which corresponded, at both autopsy and histologic examination, to a 75% narrowing of the same vessel. Acute thrombosis was suspected on PMCTA in 13 of the 14 cases detected at autopsy: in 11 cases non-perfused segments were observed and in the other 2 cases partial occlusions in non-calcified coro-

nary arteries were found. The histologic examination further revealed that seven cases were related to plaque rupture, while six to plaque erosion. The detection of infarction signs was better assessed by histology, while on PMCTA they could only be suspected in five cases where a pathological enhancement—namely an increased enhancement in the subendocardial layer during the venous and dynamic phase—was observed in regions correlating to the localization of the infarcted areas.

The ability of PMCTA to detect critical stenoses, compared to autopsy and histology, was assessed by Singh et al. [7] and Morgan et al. [14]. The first work [7] was carried out on 37 subjects, out of which a total of 158 coronary sections were obtained and included. Critical stenoses were detected in 20 coronary arteries on PMCTA and in 13 coronary arteries in histopathological examination, with a coincidence between the two methods in 8 cases. Histology further revealed changes varying from normal vessels to severe complicated atherosclerotic lesions with no or minimal patency. In 91% of cases, no significant stenoses were detected. Compared to the histopathologic examination, PMCTA showed a 61.5% sensitivity and a 91.7% specificity, with a positive predictive value (PPV) of 40% and a negative predictive value (NPV) of 96.4%. Similar results were obtained by Morgan et al. [14], who carried out their evaluations on 5 cases of sudden unexplained death, out of which 25 vessels and 568 histological sections were obtained. Thirty-nine areas of calcification were detected both histologically and on PMCTA, but the latter identified ten additional small calcification areas. Histologic examination, but not PMCTA, highlighted—within areas of critical stenosis—five regions with plaque hemorrhage, three with plaque dissection and five with recanalization. As for the detection of stenotic areas, the histologic examination identified 124 stenoses, from mild to critical, out of 25 total vessels, with agreement on PMCTA in 101 cases. The resulting sensitivity and specificity for the identification of critical stenosis per region were, respectively, 50% and 92%, with a PPV of 50% and a NPV of 92%. Such evidence demonstrates the ability of PMCTA to detect either calcifications and critical stenoses of the coronary vessels, although without giving any information on their functional relevance; conversely, no information on the nature of the atherosclerotic plaques—whether eroded or ruptured—can be obtained by this approach. As for the presence of thrombi, the latter cannot be directly detected but can only be suspected in cases of non-perfused or partially occluded coronary arteries' segments. The same results emerge from Turillazzi et al.'s [16] work, who investigated the diagnostic contribution of Multiphase PMCTA (MPMCTA) in 10 cases of sudden cardiovascular death, compared to unenhanced MPMCT and autopsy, only 2 of which revealed related to CAD and/or myocardial infarction. In one case—in which the cause of death was stated as cardiac tamponade following rupture of the free left ventricular wall associated with transmural infarct due to occlusive thrombosis of the left circumflex artery (LCX)—unenhanced MPMCT detected a diffuse calcification of both the aorta and the coronary arteries, together with massive hemopericardium. MPMCTA provided a clear evidence of the rupture of the left ventricle posterior wall and pericardial clot but, in contrast with the histopathological examination, failed to identify diffuse and advanced atherosclerotic lesions, together with an occlusive thrombus, within the LCX and areas of myocardial infarction. In the second case, a correspondence was observed between the calcification of the LAD detected by unenhanced MPMCT and the thickening of the same vessel at the level of an advanced plaque detected by MPMCTA. MPMCTA further evidenced a diffuse narrowing of the initial tract of the right coronary artery, which was confirmed as an 85–90% stenosis in histological investigation. Additionally, in this case, MPMCTA failed to detect ischemic areas otherwise identified at histological investigation.

Different results were obtained by Polacco et al. [17], who proved that PMCTA can be a valid tool for the detection of signs of myocardial infarction by highlighting contrast media extravasation in the wall of the left ventricular myocardium in 7 of the 11 cases analyzed. In six of the seven cases, the extravasation area corresponded to the distribution area of the affected coronary artery (severe stenosis or anomaly) as confirmed by autopsy and histological examination, while the remaining case, where the cause of death was

calcification of the aortic valve leaflets, was labelled as false positive. In 1 out of 11 cases, PMCTA failed to detect infarcted areas which were otherwise observed on autopsic and histologic investigations (false negative); in 1 out of 11 cases, death was stated as due to calcific aortic valve stenosis, while in the remaining 2 cases, where both subjects were dead from lethal intoxication of drugs, PMCTA, autopsy and histology did not show any pathological finding. As for the study of the coronary arteries lumen, PMCTA also proved useful in the detection of stenoses and occlusions. Specifically, normal coronary vessels (no atheroma or <50% stenosis) were observed in three cases, an intermediate to significant (50–75% stenosis) obstructive atheroma was observed in one case, while severe (>75% stenosis) obstructive CAD was observed in six cases, one of which was the one where PMCTA missed the main diagnosis of myocardial infarction.

The usefulness of PMCTA compared to native PMCT, autopsy and histology was further demonstrated in two case reports from Wan et al. [21] and Lee et al. [22]. In the case reported by Wan et al. of a 53-year-old man dead for a sudden cardiac arrest, PMCT showed severe calcification of the proximal segment of the LAD and diffuse calcification of the LCX and of the right coronary artery (RCA). Such findings corresponded, on PMCTA, to a 75–100% stenosis in the middle segment of the LAD and to a 50% stenosis of LCX and RCA; in addition, such stenoses corresponded to areas of coronary atherosclerosis. The subsequent histological examination revealed acute myocardial ischemic changes—not detected on either PMCT, PMCTA and autopsy—together with atherosclerotic changes and the degrees of calcification and stenosis consistent with the PMCT and PMCTA results. A similar correlation among the findings of the same approaches is reported by Lee et al. [22], who describe the case of a 59-year-old female, bearer of stents implanted in the proximal and middle RCA who, after developing complete thrombotic occlusion of the entire RCA, underwent an aortic hemi-arch replacement and a bypass graft using saphenous vein; despite the intervention, the patient died seven days later. In the suspicion of an acute myocardial infarction caused by thrombotic occlusion of the RCA associated with iatrogenic dissection during coronary intervention, a PMCTA was performed prior to autopsy, which showed aortic dissection with a contrast-filled true lumen and thrombosed false lumen, together with complete thrombosis of the distal segment of the RCA and complete occlusion of both implanted stents. PMCTA further showed a transmural contrast defect involving the areas of the left ventricular myocardium corresponding to the RCA territory. On gross pathological examination the stents, which revealed implanted in the false lumen of the proximal and middle segments of the RCA, showed a complete thrombotic occlusion which caused the collapse of the true lumen of the involved segments and thrombosis of the remaining true lumen of the distal segment. Macroscopic and microscopic examinations of the myocardium confirmed the presence of an infarcted area within the RCA territory.

In light of the present results, the recent implementation of angiography to conventional PMCT figures as a useful approach in the identification of coronary lesions suggestive for a cardiac cause of death, with particular reference to calcifications/atherosclerotic plaque-related stenoses and thrombosis [2–4]. Nonetheless, both the incapability to define the hemodynamic significance of the identified coronary stenoses and the possible misinterpretations due to *postmortem* changes, do not allow a sure diagnosis of CAD-related death based on PMCTA alone. The latter should thus be considered a complementary approach, rather than an alternative to autopsy, useful not only in enhancing the diagnostic accuracy of the autopsy, but also in guiding the sampling for the subsequent histological examination [2,22]. Moreover, since PMCTA does not allow the identification of plaque erosion/hemorrhage/rupture or recanalization either, histology still remains the gold standard for the diagnosis of CAD as the cause of death [14]. As for the identification of infarcted areas, although specific contrast media can highlight areas of pathologic enhancement as an indirect sign of myocardial damage [21], PMCTA is not considered suitable for the diagnosis of myocardial infarction compared to autopsy and histology.

**Table 1.** Main PMCT and PMCTA findings and comparison to autopsy and histology findings from the reviewed works.

| Author | Cases Analyzed | Main Findings |
|---|---|---|
| Michaud et al. [2] | 23 | - Calcifications detected in 18/23 cases on both PMCT and PMCTA, 12 of which correspond to thrombosis at autopsy;<br>- In total, 50–75% stenosis detected in 1/23 case on PMCTA, corresponding to an eroded plaque-related thrombosis at histology;<br>- In total, 75% stenosis of the LAD detected in 1/23 case on both PMCT and PMCTA, confirmed at both autopsy and histology;<br>- Thrombosis detected in 14/23 cases at autopsy, corresponding—on PMCTA—to 11 non-perfused segments and 2 partial occlusions in non-calcified coronary arteries. |
| Singh et al. [7] | 37<br>(158 sections) | Critical stenosis:<br>- In total, 20 cases on PMCTA;<br>- In total, 13 cases at histology;<br>- Agreement in 8 cases.<br>Sensitivity 61.5%; Specificity 91.7%; PPV 40%; NPV 96.4%. |
| Morgan et al. [14] | 5<br>(25 vessels;<br>568 sections) | - Calcifications detected in 49 sections on PMCTA, with agreement at histology in 39 cases;<br>- Mild to critical stenosis detected in 124 sections at histology, with agreement on PMCTA in 101 cases.<br>Sensitivity 50%; Specificity 92%; PPV 50%; NPV 92%.<br>Further evidence, at histology, of: plaque hemorrhage in 5 sections; plaque dissection in 3 sections; and recanalization in 5 sections. |
| Turillazzi et al. [16] | 2 | Case 1:<br>- LCx calcification detected on MPMCT;<br>- LCX thrombosis and advanced atherosclerotic plaque detected at histology but not on MPMCTA;<br>- Myocardial infarction detected at histology but not on PMCTA.<br>Case 2:<br>- LAD calcification detected on MPMCT, corresponding to wall thickening on MPMCTA and atherosclerotic plaque at histology;<br>- RCA diffuse narrowing detected on MPMCTA;<br>- Myocardial infarction detected at histology but not on PMCTA. |
| Polacco et al. [17] | 11 | - Contrast media extravasation in the wall of the left ventricular myocardium detected on PMCTA in 7/11 cases, 6 of which correspond to the distribution area of an affected (stenotic or abnormal) coronary artery; 1/7 case, false positive;<br>- Infarcted area detected in 1/11 case at both autopsy and histology but not on PMCTA (false negative). |
| Wan et al. [21] | 1 | - Severe calcification of the LAD on PMCT, corresponding to a 50–75% stenosis on PMCTA and to coronary atherosclerosis at histology;<br>- Diffuse calcification of LCx and RCA on PMCT, corresponding to a 50% stenosis on PMCTA and to coronary atherosclerosis at histology;<br>- Acute myocardial ischemic changes detected at histology but not at autopsy, PMCT or PMCTA. |
| Lee et al. [22] | 1 | - Thrombosis of the distal segment of the RCA detected on PMCTA;<br>- Transmural contrast defect involving the RCA territory, confirmed as infarcted area at both autopsy and histology. |

## 3. Role of PMMR in the Diagnosis of Myocardial Infarction as Causes of Death

Due to its superiority in the study of soft tissues, within a *postmortem* context PMMR has proved better than PMCTA in the detection of myocardial infarction [3–5,23], also allowing the estimation of the infarction age, as in the work by Jackowski et al. [24] who, for

the first time, engaged in aging myocardial infarction areas by comparing the findings of unenhanced PMMR in $T_1$, $T_2$, STIR and FLAIR sequences, autopsy and histology on eight cases selected from the Virtopsy project. PMMR, autopsy and histology failed to identify signs of peracute ischemia in two cases in which myocardial infarction was stated as the cause of death following the identification of a fresh coronary occlusion at autopsy. In four cases where acute ischemia was identified, a correspondence was found between the core necrotic areas histologically detected and the signal reduction of the same areas on $T_2$-weighted, STIR and FLAIR sequences. Subacute infarcted regions were highlighted in other four cases as characterized by fibroblasts replacing the necrotic fibers and ingrowing vessels in histology, and increased signals on $T_2$-weighted, STIR, and FLAIR images. As for chronic infarction, it was identified as definite collagen deposits in histology and decreasing signal on—in succession—$T_2$, STIR, FLAIR and $T_1$ sequences. Comparable results were obtained in another work from Jackowski et al. [25] where they carried out on 19 lesions analyzed from 16 cases. In this second work, STIR and FLAIR sequences were replaced by PD-weighted sequences. Additionally, in this case, a good correlation was observed between PMMR, autopsy and histology in aging acute, subacute and chronic infarction areas in 13/19 lesions analyzed: specifically, $T_2$-weighted sequences showed acute infarction as a hypointense center with hyperintense margins related to oedema in the outer zone, while subacute infarction displayed hyperintense alterations within the affected myocardium; chronic infarction showed a broad loss of signal in all sequences. The same conclusions were reached by Zech et al. [26] and Schwendener et al. [27] by relying on a quantitative approach using, respectively, a 3-T and a 1.5-T MR system and comparing the findings to those obtained on autopsy and histology (Table 2).

Although PMMR has proved more suitable for the study of myocardial tissue alterations within infarcted areas, Ruder et al. [28] relied on such approach in order to assess its value in the identification of CAD by evaluating the Chemical Shift Artifact (CSA), a MR artifact occurring at the interface between fat and water which appears as light and dark bands on opposite sides of a structure. As such, the CSA should only occur in patent vessels, while the presence of a significant stenosis should either prevent or alter it. On such basis, the authors evaluated, on *postmortem* cardiac MR, whether the CSA could identify patent vessel segments in the coronary arteries from 30 cases analyzed. Of the 300 vessel segments assessed, autopsy revealed the presence of stenosis in 74 segments, in 35 of which it appeared critical. On MR, CSA was observed in 112/300 segments while paired dark bands (PDB)—indicative of pathologic changes—were identified 58/300 segments. Neither CSA nor PDB were detected in 71/300 vessel segments. The comparison of MR and autopsy findings revealed the exclusive occurrence of CSA in vessels with no or no significant stenosis (112/112), but never in vessels with significant stenosis (0/112). PDB were detected in 37 vessel segments with pathologic changes and in 21 vessel segments with no pathology, and were overall more often present in vessels with significant stenosis than in vessels with insignificant stenosis.

Although more useful in the detection of infarcted areas, as well as PMCTA also cardiac PMMR—due to the resulting unspecific findings—is mostly useful in guiding further autopsic and histologic investigations rather than providing a definite diagnosis. Nonetheless, despite its value as a complementary approach, PMMR is not yet established in *postmortem* investigations due to several limitations, including the high costs, long examination times, complexity, need for trained equips and, last but not least, the absence of dedicated protocols due to the limited number of studies carried out [3,5].

**Table 2.** Main PMMR findings and comparison to autopsy and histology findings from the reviewed works.

| Author | Cases Analyzed | Main Findings |
|---|---|---|
| Jackowski et al. [24] | 8 | - Peracute ischemia identified in 2/8 cases following the detection of a fresh coronary occlusion at autopsy; no tissue alterations detected on PMMR, autopsy or histology;<br>- Acute ischemia identified in 4/8 cases: signal reduction on $T_2$, STIR and FLAIR sequences corresponding to core necrotic areas at histology;<br>- Subacute infarction detected in 4/8 cases: increased signal on $T_2$, STIR and FLAIR sequences corresponding to ingrowing vessels and fibroblasts replacing necrotic fibers at histology;<br>- Chronic infarction detected in 4/8 cases: decreasing signal from $T_2$ to STIR to FLAIR to $T_1$ sequences, corresponding to definite collagen deposits at histology. |
| Jackowski et al. [25] | 16 | In total, 19 myocardial lesions identified and age staged on PMMR as follows:<br>- Peracute myocardial lesions were detected on PMMR but not at autopsy or histology; in such cases, histology revealed severe coronary stenosis;<br>- Acute infarction showed central hypointensity on $T_2$ sequences corresponding to core necrotic areas at histology + marginal hyperintensity on $T_2$ sequences corresponding to peripheral oedema at histology;<br>- Subacute infarction showed hyperintensity on $T_2$ sequences corresponding to the affected myocardial areas at histology;<br>- Chronic infarction showed a broad loss of signal corresponding to the affected myocardial areas at histology. |
| Zech et al. [26] | 16 lesions | PMMR findings on $T_1$, $T_2$ and PD sequences:<br>- In total, 8 acute lesions: central hypointensity on $T_2$ sequences corresponding to core necrotic areas at histology + marginal hyperintensity on $T_2$ sequences corresponding to perifocal oedema at histology;<br>- In total, 8 subacute lesions: hyperintensity on $T_2$ sequences corresponding to the affected myocardial areas at histology;<br>- In total, 6 chronic lesions: broad loss of signal corresponding to the affected myocardial areas at histology. |
| Schwendener et al. [27] | 80 cases | - In total, 73 focal myocardial signal alterations out of 49/80 cases, histologically corresponding to:<br>- Early acute ischaemia in 39/73 lesions;<br>- Acute infarction in 14/73 lesions;<br>- Subacute infarction in 10/73 lesions;<br>- Chronic infarction in 10/73 lesions. |
| Ruder et al. [28] | 30 (300 segments) | - Chemical Shift Artifact (CSA) detected in 112/300 segments, all corresponding to no or no significant stenosis at autopsy;<br>- Coronary stenosis detected in 74/300 segments at autopsy, corresponding to the absence of CSA in 71/74 segments analyzed. |

## 4. Role of PMMR in the Diagnosis of Cardiomyopathies as Causes of Death

Due to its capability to provide useful information concerning both cardiac morphology and cardiac quantitative parameters (e.g., atrial and ventricular mass, volumes and dimensions), within the last few years Cardiac PMMR (PMCMR) has proved a useful approach for the study of cardiomyopathies [29] (Table 3).

With regard to Hypertrophic Cardiomyopathy (HCM), the well-known diagnostic role of cardiac MR in a clinical setting has led several forensic researchers to engage in the assessment of its value also within a post mortal setting. The resulting evidence emerged from a work from Jackowski et al. [30] showed that ventricle hypertrophy and dilation can be evaluated on both a short axis view and a horizontal and vertical long axis view by manually tracing the endocardial and epicardial contours at the workstation; in addition, the calculation of the heart weight—useful for distinguishing normal sized from

hypertrophied hearts—can be provided by either multiplying the myocardial volume for the coefficient $1.05$ g/cm$^3$, as stated by the authors—assumed density of the myocardium—or relying on a commercially available mass analyzing software. The elaboration of such morphovolumetric data have important applicative implications concerning the analysis of primary and secondary hypertrophy, since they provide precise and detailed information on the cardiac wall thickening [31–33]. Other morphological features detected by PMMR include intramyocardial bridge, ventricular crypts and anomalies of papillary muscles and sub-mitral apparatus, which support the diagnosis of HCM [29].

In their work, Acquaro et al. [34] proposed a new method for the diagnosis of HCM by PMCMR despite myocardial rigor mortis. To this aim, firstly they performed Cardiac MR (CMR) in a total of 111 living patients (49 with HCM; 30 with secondary cardiac hypertrophy; 32 healthy control patients) simulating rigor mortis as the mid-diastolic cardiac phase of CMR cine-image, thus defining new morphological parameters for the diagnosis of HCM based on left ventricular mass, wall thickness (WT) mean and standard deviation (SD), maximal and minimal WT and their differences. Such parameters were then validated at PMCMR following the evaluation of a total of 28 hearts, 8 of which obtained from HCM patients, 10 from patients with CAD and 10 from patients with non-cardiac death. In the in vivo study, the SD of WT was significantly higher in HCM than in healthy controls and in non-HCM hypertrophy patients, with a cut-off $> 2,4$ corresponding to the highest AUC. Additionally, at PMCMR, the SD of WT was significantly higher in HCM than in CAD ($p = 0.003$) and in non-cardiac death ($p = 0.0009$), and the 2,4 threshold established in the in vivo study was able as well to detect all of the eight patients with HCM, but none of those with CAD or non-cardiac death. In light of such results, the authors concluded that a SD of WT with a 2,4 threshold allows HCM identification at PMCMR.

The potential of PMCMR has also been evaluated in the diagnosis of the Arrhythmogenic Right Ventricular Cardiomyopathy (ARVC) since such investigation provides a clear differentiation of fatty and muscular tissues by $T_1$-weighted TSE imaging—fat corresponding to a high signal intensity, whereas myocardium corresponds to an intermediate signal intensity [29,30,35]. Nonetheless, even though such method is effective in tissue discrimination, the only evidence of a fatty infiltration cannot be considered sufficient for the diagnosis of ARVC disease, being common to other different myocardial diseases as well as also described in healthy adults [29].

The potential of PMCMR in the diagnosis of primary and secondary cardiomyopathies has also been evaluated following the implementation of Late Gadolinium Enhancement (LGE), as discussed in an interesting review from Hashimura et al. [36], knowing that LGE areas histologically correspond to interstitial expansion due to fibrosis, abnormal protein deposition, inflammatory infiltrates, granulomas and cardiomyocytes necrosis. Specifically, in histology, LGE corresponded to (i) replacement fibrosis and fibrofatty change in both hypertrophic and dilated cardiomyopathies, (ii) fibrofatty replacement in arrhythmogenic right ventricular cardiomyopathy, (iii) epithelioid granuloma and fibrosis in cardiac sarcoidosis, (iv) inflammatory infiltrates, cardiomyocytes necrosis and replacement fibrosis in giant cell myocarditis and (v) amyloid deposition, fibrosis and coagulative cardiomyocytic necrosis in cardiac amyloidosis.

**Table 3.** Main PMMR findings concerning cardiomyopathies from the reviewed works.

| Author | Cases Analyzed | Main Findings |
|---|---|---|
| Jackowski et al. [30] | 80 | - Ventricle hypertrophy and dilation evaluated on both a short axis view and a horizontal and vertical long axis view by manually tracing the endocardial and epicardial contours at the workstation;<br>- Calculation of the heart weight, and subsequent distinction of normal sized from hypertrophied hearts, obtained by either multiplying the myocardial volume for the coefficient 1.05 g/cm3—assumed density of the myocardium—or relying on a commercially available mass analyzing software. |
| Aquaro et al. [34] | Cardiac MR in 111 living patients (49 with HCM; 30 with secondary cardiac hypertrophy; 32 healthy controls). PMCMR on 28 hearts (8 from HCM patients; 10 from CAD patients; 10 from patients with non-cardiac death). | In vivo study:<br>- Simulation of rigor mortis as mid-diastolic cardiac phase of CMR cine image;<br>- Definition of new morphological parameters for the diagnosis of HCM (left ventricular mass; wall thickness (WT) mean and standard deviation (SD); maximal and minimal wall thickness and their differences);<br>- SD of WT significantly higher in HCM than in healthy controls and in non-HCM hypertrophy patients;<br>- Cut-off > 2,4 corresponding to the highest AUC.<br>Test of the new parameters on PMCMR:<br>- SD of WT significantly higher in HCM than in CAD ($p = 0.003$) and in non-cardiac death ($p = 0.0009$);<br>- Cut-off > 2,4 established in the in vivo study, able as well to detect all of the 8 patients with HCM, but none of those with CAD or non-cardiac death. |
| Mondello et al. [35] | 1 | PMMR evidence of Arrhythmogenic Right Ventricular Dysplasia, further confirmed at autopsy and histology. |
| Hashimura et al. [36] | - | Late Gadolinium Enhancement (LGE) areas—corresponding to interstitial expansion due to fibrosis, abnormal protein deposition, inflammatory infiltrates, granulomas and cardiomyocytes necrosis—histologically relates to:<br>- Replacement fibrosis and fibrofatty change in both hypertrophic and dilated cardiomyopathies;<br>- Fibrofatty replacement in arrhythmogenic right ventricular cardiomyopathy;<br>- Epithelioid granuloma and fibrosis in cardiac sarcoidosis;<br>- Inflammatory infiltrates, cardiomyocytes necrosis and replacement fibrosis in giant cell myocarditis;<br>- Amyloid deposition, fibrosis and coagulative cardiomyocytic necrosis in cardiac amyloidosis. |

## 5. Advances and Limitations

The introduction of CT techniques within a forensic context is to date considered an advantage in selected cases. Overall, the use of PMCT has proved useful for a better definition of traumatic causes of death, especially for its capability to detect injuries which could not otherwise be evidenced on a traditional autopsy. In the context of cardiovascular diseases, PMCT is undoubtedly useful in the identification of lesions such as hemopericardium or aortic ruptures, as well as in the localization of cardiovascular devices and definition of the heart size. Nonetheless, except for the detection of calcifications, the impossibility to yield clear information concerning the coronary arteries due to the lack of an active circulation, has led to the introduction of PMCTA for its high ability to enhance the cardiovascular system visualization. Such an approach has allowed the evaluation of a detailed distribution pattern of the epicardial coronary arteries, as well as the assessment of the extent and localization of atherosclerotic plaque-related stenoses and thrombosis, suggestive for a cardiac cause of death [7,14,16].

On the other hand, the hemodynamic significance of the identified coronary lesions cannot be defined, and misinterpretations due to *postmortem* changes or radiologic artifacts are a concrete possibility. For example, *postmortem* blood clots (also known as "cruor mortis") may be misinterpreted as intravascular thrombi responsible for myocardial infarction, which can be avoided by observing whether the clots are non-adherent to the vessel (in such case, the contrast medium surrounds the clots). Contrast layering and inhomogeneous opacification of the vessels can mimic a stenosis in the coronary artery, for whose differentiation from a *postmortem* artifact all three phases have to be analyzed on PMCTA. Alongside the issues related to radiological artifacts, another problem concerns the lack of trained personnel. Moreover, since PMCTA does not allow the identification of either infarcted areas and plaque erosion/hemorrhage/rupture or recanalization, a sure diagnosis of CAD-related death based on PMCTA alone is not possible, and histology still remains the gold standard. As such, PMCTA should be considered a complementary approach to autopsy, with the additional advantage of guiding the sampling for subsequent histological examinations.

Compared to PMCTA, PMMR can provide information on the soft tissues of the cardiovascular system, on cardiac morphology and on cardiac quantitative parameters, thus making it a better approach for the detection of myocardial ischemia/infarction and cardiomyopathies [24–27,30–33]. Nonetheless, in this case several artefacts can lead to misinterpretations, such as: chemical shift artefact, due to frequency difference between adjacent fat and water; susceptibility artefact, due to other differences between tissues; and magnetic artefacts due to an overlapping of slices which can lead to image deformation and loss of information during the acquisition [28,29]. As such, PMMR alone is not useful in providing a definite diagnosis of cardiac death. Although PMMR could be mostly useful in guiding further autopsic and histologic investigations, it is not yet established in a *postmortem* context due to several limitations: the procedure is too expensive; scanning times are long; trained personnel are requested; and no dedicated protocols are available. Considering the advantages and disadvantages of both procedures, even though a definite diagnosis of cardiac death cannot always be provided, the possibility to guide further investigations makes PMCTA a more feasible approach, also due to the relatively low maintenance costs and fast scanning times. In fact, even if both autopsy and histological analysis are fundamental for the *postmortem* assessment of a cardiac disease, PMCT is useful for forensic pathologists, being a strategic guide during autopsy. Moreover, if compared to heart gross examination, the PMCT acquired data provide overall and detailed information on the entire coronary tree and myocardial tissue. The acquired data can be re-used and the images can be re-analyzed many times supporting the pathologist during the judicial activities.

The implementation of forensic departments with dedicated PMCT instruments and trained personnel is thus recommended, focusing on the importance of the collaboration between trained radiologists and forensic pathologists.

Finally, it must be highlighted that the *postmortem* radiological analysis performed for cardiac disease evaluation could integrate the clinical and radiological knowledge offering findings which contribute to the improvement of the medical research on radiological diagnosis of heart disease with an impact on public health.

## 6. Conclusions

The implementation of forensic investigations with radiological approaches is fast emerging as an alternative to autopsy in a number of selected cases such as traumatic deaths. In such a context, in an attempt to identify CAD and myocardial infarction as causes of death with a minimally invasive approach, several PMCTA protocols have been developed in order to study lumen, course and wall pathology of the coronary vessels. Despite the undeniable aid provided by PMCTA in either the detection of calcifications and atherosclerotic plaque-related stenoses and thrombosis suspicion within non-perfused or partially occluded coronary vessels, autopsy and histology still remain the gold standard

for a definite diagnosis in cases of cardiac death [1,3–5,16]. As for the detection of infarcted areas, although the use of specific contrast media can provide indirect evidence of ischemic necrosis within areas of pathological myocardial enhancement [4], PMMR has proved more suitable for such purpose due to its best ability to detect soft tissue changes, also providing the aging of the infarcted areas depending on the sequences analyzed. Nonetheless, as well as PMCTA, PMMR alone is not sufficient to relate with certainty the cause of death to myocardial infarction or even cardiomyopathies.

Unfortunately, there are only few works produced on the topic discussed here, and the casuistries analyzed are not wide enough. As a consequence, although the undeniable utility of such approaches, the main limit of the present work is represented by the impossibility to establish their effective value in the diagnosis of cardiac death due to the lack of adequate positive and negative predictive values. In light of this, both PMCTA and PMMR can only be addressed as useful complementary approaches, aimed at guiding further forensic investigations in cases suggestive of cardiac death. In such cases, a multidisciplinary approach—further comprehensive of clinical history evaluation, analysis of cardiac biomarkers and/or genetic investigations, whenever available—is also advisable [5,17].

**Author Contributions:** Conceptualization, C.M. and E.V.S.; methodology, C.S. and G.B.; investigation, L.C., P.G., D.S. and F.C.; writing—original draft preparation, C.S. and C.M.; writing—review and editing, A.A. and E.V.S.; supervision, E.V.S. and A.A. All authors equally contributed to the manuscript. All authors have read and agreed to the published version of the manuscript.

**Funding:** This research has received no external funding.

**Institutional Review Board Statement:** Not applicable.

**Informed Consent Statement:** Not applicable.

**Data Availability Statement:** Not applicable.

**Conflicts of Interest:** The authors declare no conflict of interest.

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
