# Peer review of "State of the Art on the Role of Postmortem Computed Tomography Angiography and Magnetic Resonance Imaging in the Diagnosis of Cardiac Causes of Death: A Narrative Review"

_tomography, doi:10.3390/tomography8020077_

Round 1

Reviewer 1 Report

This review summarized the role of Post Mortem Computed Tomog- 2
raphy Angiography and Magnetic Resonance Imaging in diagnosis of cardiac cause of death. Compared with standard autopsy and histology, they found both PMCTA and PMMR could work as useful complementary approaches and provide supporting evidence for diagnosis. 

The manuscript has a clear introduction and lists the reviewed article in the tables which are easy to follow. Here are some minor suggestions: 

  1. There is no information about how to select the reviewed articles and bascially, even there is not that much exist, what's the keyworks for article searching? And is there any exclusion?
  2. Also, there is no limination in the discussion part, please add some statements to discuss the limitations of this review.

Author Response

Reply to Reviewer 1

Thank you for your kind suggestions, in accordance to which the proper corrections to the manuscript have been provided.

  1. There is no information about how to select the reviewed articles and basically, even there is not that much exist, what’s the keywords for article searching? And is there any exclusion?

Unlike systematic reviews, narrative reviews don’t rely on acknowledged guidelines; for this reason, a description of the research strategy is not routinely requested. Nonetheless, in order to meet your request, a brief overview has been provided at the end of the introduction paragraph, highlighted in italics, bold and red colour.

  1. There is no limitation in the discussion part, please add some statements to discuss the limitations of this review

As suggested, a brief statement has been added at the end of the paragraph “6. Conclusion”, highlighted in italics, bold and red colour.

Reviewer 2 Report

In this manuscript, Stassi et al. have reviewed the applications of post-mortem CT, CTA and MRI in the diagnosis of cardiac causes of death. Cardiac disease is a major cause of unexpected deaths worldwide, this article will be of interest. I have the following minor concerns/suggestions:

  1. Authors should include the search strategy, key words used, and search databases used for the collection of literature data for this manuscript.
  2. A paragraph should be added before the Conclusions, briefly discussing advantages and limitations (of PMCT/CTA and PMMR), and future directions in terms of feasibility and logistics of imaging deceased patients in the CT/MRI scanners which are generally used for routine diagnostic imaging.

Author Response

Reviewer 2

Thank you for your kind suggestions, in accordance to which the proper corrections to the manuscript have been provided.

  1. Authors should include the search strategy, keywords used, and search databases used for the collection of literature data for this manuscript

Unlike systematic reviews, narrative reviews don’t rely on acknowledged guidelines; for this reason, a description of the research strategy is not routinely requested. Nonetheless, in order to meet your request, a brief overview has been provided at the end of the introduction paragraph, highlighted in italics, bold and red colour.

  1. A paragraph should be added before the Conclusions, briefly discussing advantages and limitations (of PMCT/CTA and PMMR), and future directions in terms of feasibility and logistics of imaging deceased patients in the CT/MRI scanners which are generally used for routine diagnostic imaging

As suggested, a new paragraph entitled “5. Advantages and Disadvantages” has been added, highlighted in italics, bold and green colour.

Reviewer 3 Report

The authors provide a thorough narrative review of the use of post mortem CT and MR imaging modalities as an adjunct to autopsy and histological examination of the heart in determining cardiac causes of death. Overall, the manuscript is comprehensive and well structured. There are some English grammatical issues and a need to reword some sentences in order to help the readability but this is not a major issue and can be easily improved.

Minor grammatical changes:

  • Line 41 - "the rely on a minimally invasive ..." (please reword for clarity)
  • Line 52 - "Despite such advances, a major limit was represented ..." (please reword for clarity)
  • Line 54 - "causes of death due to the lack of and active approaches" (delete the word and)
  • Line 77, 194, 212 - please change post mortal to post mortem
  • Line 99 - "histology still revealed the gold standard since infarction.." - (please reword for clarity, this sounds like histology was defending itself against PMCTA as the gold standard for detection)
  • Line 109 - "In 91% cases any significant stenosis" (please reword for clarity)
  • Line 122 - "Such evidences testify" (Please reword for clarity)
  • Line 225 - "decreasing signal from T2 to stir to flair to T1" (this is confusing and when using acronyms for pulse sequences they should be capitalized)
  • Lines 332 - "contrast media has revealed able to provide" (please reword for clarity)

My major concern with this article is that the major conclusion is that CT and MR imaging is not sufficient alone to determine with certainty the cause of death to myocardial infarction or cardiomyopathies and that at best is a complementary piece to the standard autopsy and histopathological examination. I can see how the review sheds light on the adjunctive potential of imaging modalities, I struggle to see how this approach will ever be widely used as in many cases will be cost prohibitive. Therefore, I believe it is necessary for the authors to provide a separate section that details the usefulness of costly imaging modalities post mortem from a public health perspective as well as plainly discuss the limitations to the use of post mortem imaging modalities.

Author Response

Reviewer 3

Thank you for your kind suggestions, in accordance to which the proper corrections to the manuscript have been provided.

  1. Minor grammatical changes:

Line 41 – “the rely on a minimally invasive …” (please reword for clarity);

Line 52 – “Despite such advances, a major limit was represented …” (please reword for clarity);

Line 54 – “causes of death due to the lack of and active approaches” (delete the word and);

Lines 77, 194, 212 - please change post mortal to post mortem;

Line 99 – “histology still revealed the gold standard since infarction …” - (please reword for clarity, this sounds like histology was defending itself against PMCTA as the gold standard for detection);

Line 109 – “In 91% cases any significant stenosis …” (please reword for clarity);

Line 122 – “Such evidences testify …” (please reword for clarity);

Line 225 – “decreasing signal from T2 to stir to flair to T1” (this is confusing and when using acronyms for pulse sequences they should be capitalized);

Line 332 – “contrast media has revealed able to provide …” (please reword for clarity)

All listed grammatical changes have been applied according to your suggestions. For a faster check, they have been highlighted in italics, bold and violet colour.

  1. My major concern with this article is that the major conclusion is that CT and MR imaging is not sufficient alone to determine with certainty the cause of death to myocardial infarction or cardiomyopathies and that at best is a complementary piece to the standard autopsy and histopathological examination. I can see how the review sheds light on the adjunctive potential of imaging modalities, I struggle to see how this approach will ever be widely used as in many cases will be cost prohibitive. Therefore, I believe it is necessary for the authors to provide a separate section that details the usefulness of costly imaging modalities post mortem from a public health perspective as well as plainly discuss the limitations to the use of post mortem imaging modalities

As suggested, a new paragraph entitled “5. Advantages and Disadvantages” has been added, highlighted in italics, bold and green colour.

Round 2

Reviewer 3 Report

Thank you for your detailed and easily identifiable recommended changes to the manuscript. I have a couple minor revisions:

  • Line 329 - "The rely on PMCT has overall proved useful for a better defini- tion of traumatic causes of death" (not sure what you mean by "The rely"), please reword
  • I appreciate the inclusion of an advantages and disadvantages section, however, it still does not address the public health importance of utilizing these imaging modalities to help with autopsy and histopathology in determining the cause of death. Also, CT and MRI are expensive modalities that require trained personnel to perform the exam and read the exam. My overlying concern on the initial report was why use these techniques if autopsy is the gold standard, or better yet, what public health  advantage (how can these modalities improve public health given they are an expensive complement to autopsy/histology)? Please answer this question in section 5.

Author Response

Thank you for your detailed and easily identifiable recommended changes to the manuscript. I have a couple minor revisions:

  • Line 329 - "The rely on PMCT has overall proved useful for a better defini- tion of traumatic causes of death" (not sure what you mean by "The rely"), please reword

The change was performed and is highlighted in italics, bold and blue colour.

  • I appreciate the inclusion of an advantages and disadvantages section, however, it still does not address the public health importance of utilizing these imaging modalities to help with autopsy and histopathology in determining the cause of death. Also, CT and MRI are expensive modalities that require trained personnel to perform the exam and read the exam. My overlying concern on the initial report was why use these techniques if autopsy is the gold standard, or better yet, what public health  advantage (how can these modalities improve public health given they are an expensive complement to autopsy/histology)? Please answer this question in section 5.

As suggested, some consideration were added in section 5, highlighted in italics, bold and blue colour.